# The Acari Hypothesis, III: Atopic Dermatitis

**DOI:** 10.3390/pathogens11101083

**Published:** 2022-09-23

**Authors:** Andrew C. Retzinger, Gregory S. Retzinger

**Affiliations:** 1Department of Emergency Medicine, Camden Clark Medical Center, West Virginia University, Parkersburg, WV 26101, USA; 2Department of Pathology, Feinberg School of Medicine, Northwestern University, Chicago, IL 60611, USA

**Keywords:** Acari Hypothesis, atopy, atopic dermatitis, mites, IgE, *Demodex*, *Dermatophagoides*, *Malassezia*, heat shock proteins

## Abstract

Atopic dermatitis is a chronic relapsing dermatopathology involving IgE against allergenic materials present on mammalian epithelial surfaces. Allergens are as diverse as pet danders, and polypeptides expressed by microbes of the mammalian microbiome, e.g., *Malassezia* spp. The Acari Hypothesis posits that the mammalian innate immune system utilizes pathogen-bound acarian immune effectors to protect against the vectorial threat posed by mites and ticks. Per The Hypothesis, IgE-mediated allergic disease is a specious consequence of the pairing of acarian gastrointestinal materials, e.g., allergenic foodstuffs, with acarian innate immune effectors that have interspecies operability. In keeping with The Hypothesis, the IgE profile of atopic patients should include both anti-acarian antibodies and specious antibodies responsible for specific allergy. Further, the profile should inform on the diet and/or environment of the acarian vector. In this regard, the prevalence of *Demodex* and *Dermatophagoides* on the skin of persons suffering from atopic dermatitis is increased. Importantly, the diets of these mites correspond well with the allergens of affected patients. In this report, roles for these specific acarians in the pathogenesis of atopic dermatitis are proposed and elaborated.

## 1. Introduction

The Acari Hypothesis posits IgE-mediated immunity is intended primarily to protect mammals against pathogen-bearing acarians, i.e., mites and ticks. Further, it theorizes the means by which such adaptive responses occur [1,2]. Importantly, The Hypothesis informs that most allergens derive from molecular species well-represented within the gastrointestinal contents of acarians, i.e., tropomyosin, profilin, EF-hand domains, etc. [3,4,5]. When structurally similar materials from acarian foodstuffs complex with adjuvant-active innate immune effectors present in acarian gastrointestinal contents, e.g., tick saliva or mite stool, the mammalian immune system misconstrues the complexes as acarian representations, targeting them adaptively with an anti-acarian response.

The Hypothesis expounds the benefit of IgE-mediated immunity as an intended anti-vector defense whilst anticipating the potential for immunopathology, a consequence of specious IgE against non-vector materials. The original proposal focused on anaphylaxis, the prototypical early phase allergic reaction, because that reaction is suited in both scale and locale to removal of invasive ectoparasites [1]. The late phase of allergic inflammation, which is characterized by the persistent release of chemical mediators from resident mast cells, infiltration of T_H_2 cells, and infiltration and degranulation of eosinophils [6], seems equally suited to ectoparasite removal. IgE-activated mast cells release chemokines that facilitate chemotaxis of late phase immune effectors that drive the T_H_2 response [7]. It is therefore reasonable to regard late phase allergy as a continuation of processes initiated by early phase events.

The late phase of allergy also aligns teleologically with an anti-acarian objective. If physical means do not remove the ectoparasite in rapid fashion, the local environment becomes toxic, thereby either eliminating the acarian or influencing its migration. The late phase of allergy, then, represents local escalation of the allergic process. Inasmuch as acarian-borne pathogens can kill mammalian hosts [8], reparatory injury to infected tissue is entirely appropriate.

The character of allergic inflammation depends upon the duration of allergen exposure: transient exposure elicits an early phase response; prolonged exposure includes a late phase response; and unremitting exposure yields chronic inflammation. In the case of foodstuffs and certain environmental allergens, e.g., peanuts and bee venom, removal of a limited amount of exogenous allergen is uncomplicated and the manifestations of exposure are singular and transient. Similarly, seasonal exposures to other exogenous allergens, e.g., pollens and/or mold spores, yield seasonal manifestations. However, the mammalian immune system also encounters persistent microbial antigens, e.g., skin flora. And, just as transient antigens incorporate erroneously into an anti-acarian response, so, too, do persistent ones, the failed clearance of which by IgE yields chronic disease.

Atopic dermatitis (AD) is a chronic IgE-mediated inflammatory condition [9]. As was just alluded, its chronicity indicates unremitting exposure to an eliciting allergen. Because a very significant percentage of AD patients make IgE against *Malassezia* [10,11,12], these commensal organisms of normal skin flora are ideally suited as eliciting allergen. According to The Hypothesis, IgE is made against materials complexed with an adjuvant-active acarian immune effector [2]. Complexation occurs within the gastrointestinal tract of the acarian, prior to transmittance. Because the existence of IgE presupposes an acarian exposure, anti-*Malassezia* IgE must then be a consequence of the affected individual having been exposed to an acarian harboring *Malassezia* materials within its gastrointestinal tract. Thus, the responsible acarian most likely consumes fungi as a part of its diet. Given that AD patients routinely produce IgE against the danders of cats and dogs, the culpable acarians likely either inhabit or regularly approximate mammalian skin, i.e., *Demodex* and/or *Dermatophagoides*.

## 2. Atopic Dermatitis

AD is a chronic relapsing dermatopathology that in developing countries now affects 10–20% of children and 1–3% of adults [13]. As no definitive biomarker for AD yet exists, its diagnosis is based primarily upon clinical presentation and histopathological findings [14]. Clinical presentations are heterogeneous, especially regarding time course, anatomic distribution and eliciting allergen [15,16]. The condition is often difficult to distinguish from other dermatitides, especially primary acarian infestations, examples of which include scabies (*Sarcoptes scabiei*), and demodicosis (*Demodex* spp.) [17,18,19]. Histopathological evaluation of AD lesions reveals a thickened, hyperkeratotic epidermis accompanied by T_H_2-polarized inflammation [20]. Deposition of eosinophil granular proteins, including major basic protein, is extensive [21,22]. Resident mast cells and infiltrating CD4+ lymphocytes drive the response via induction of IL-4, IL-5, IL-13 and IL-31 pathways [7,23,24,25]. IL-5 stimulates eosinophilopoiesis and eosinophil activation and chemotaxis [26]. 

Despite significant understanding of the cellular effectors and the cytokine pathways involved in AD, the pathogenesis of the condition remains poorly understood. In this regard, there are two competing hypotheses. One, the ‘inside-out hypothesis’, proposes aberrant IgE sensitization prompts inflammation that compromises the integrity of the epithelial barrier. The other, the ‘outside-in hypothesis,’ proposes disruption of the epithelial barrier enables penetration of allergenic materials into the skin, resulting in immune activation [27,28]. As loss-of-function mutations in proteins that maintain epidermal barrier function are a significant risk factor for AD [29,30], the latter hypothesis is favored. That hypothesis also better explains one of the more perplexing aspects of AD, namely, atopic march. 

Atopic march refers to the sequential progression of atopic manifestations that occurs in a subset of AD patients [31,32]. Multiple studies have correlated AD with the subsequent development of allergic rhinitis, IgE-mediated food allergies, and asthma [33,34,35,36]. An array of IgE usually accompanies atopic march, especially with food allergies. Affected individuals appear to have an immune-enhancing or ‘adjuvant’ activity specific for IgE. Current thinking attributes the adjuvant activity to compromised epidermal barrier, i.e., the outside-in hypothesis asserts that sub-epidermal penetration of allergenic materials somehow enhances formation of IgE [37,38].

The Acari Hypothesis accounts for AD, atopic march, and related phenomena altogether differently. According to it, the immune system of an individual with AD is not dysregulated. Instead, that immune system is responding as intended to a vector-active acarian. Having paired with an acarian pattern recognition receptor, dietary elements—even materials from human skin flora—of the culprit mite or tick are perceived by a human recipient as acarian in nature, subsequently eliciting ‘specious’ IgE and its attending inflammation. Atopic march is then attributable to persistent or repeated infestation by a vector-active acarian, not to compromised barrier integrity/function.

## 3. IgE in Atopic Dermatitis

IgE is uniquely mammalian, and all extant mammals express it. It initiates and directs atopic inflammation, including AD [39]. In humans, IgE binds to leukocytes via high, i.e., FcεRI, and/or low, i.e., FcεRII (CD23) affinity receptors [40]. In atopic conditions, complexation of allergen and IgE bound to FcεRI drives a T_H_2-polarized response in leukocyte-specific fashion. In mast cells and basophils, allergen-induced crosslinking of bound IgE elicits release of pre-formed inflammatory mediators, e.g., histamine [7]. Dendritic cells expressing FcεRI are increased in AD patients [41,42], and dendritic cells in lesional skin of both humans and canines bind IgE [43,44]. Activation of FcεRI-expressing dendritic cells stimulates newly activated naive T_H_ cells along the T_H_2 differentiation pathway, amplifying the T_H_2 response [45]. Interestingly, Langerhans cells in AD lesional skin extend their dendrites beyond the epidermal tight junction [46], a measure eminently suited to detecting the external threat posed by ectoparasites. Although the role of CD23 in atopic inflammation is less well-defined, CD23 does seem to have critical involvement in IgE synthesis [47]. Importantly, Der p 1, an allergenic protease expressed by *D. pteronyssinus*, cleaves CD23 from the surface of human B cells [48]. Whether such cleavage facilitates or hinders mammalian immunity is unclear. Nevertheless, conservation of the protease across acarian species [49,50] argues persuasively for the existence of a special relationship between acarians and IgE. 

IgE is central to atopy and a clear link between IgE and AD exists; however, the exact role of IgE in AD remains unsettled. The reason is that ~20% of patients with AD do not have elevated levels of either total or specific IgE [51]. This, in turn, has led to categorization of clinical AD as being either ‘intrinsic’ or ‘extrinsic.’ Whereas extrinsic AD is characterized by a high IgE level and/or the presence of specific IgE, intrinsic AD is characterized by a normal IgE level and the absence of specific IgE [52]. Different endotypes may represent distinct clinical entities. Against this, a significant subset of patients with intrinsic AD eventually develops elevated levels of total and/or specific IgE [53]. The Acari Hypothesis explains endotype variation. Central to The Hypothesis is the notion that the adaptive IgE-mediated allergic response targets acarians. To be consistent with the existing paradigm of adaptive immunity, there must be for IgE a priming innate response that recognizes pathogen-associated molecular patterns or—more appropriately—vector-associated molecular patterns. In this regard, a growing body of evidence implicates chitin, the major polysaccharide of acarian exoskeletons, including those of *Demodex* spp. [54]. Chitin initiates a T_H_2 response by mobilizing the same cells and eliciting the same cytokines as IgE [55,56]. What, then, drives the generation of IgE? As proposed previously, the chief determinant is pathogen transmission by an infesting acarian [2].

Recognition of the centrality of IgE in the pathophysiology of AD has prompted many attempts to characterize the entirety of IgE allergen profiles [12,57,58,59,60,61]. A plethora of allergens have been identified, including: (1) materials derived from the epithelium of domesticated mammals, (2) materials expressed by resident microbes on mammalian skin, and (3) bioaerosols trapped within mammalian nares. Of the microbial allergens residing on mammals, ones attributable to *Malassezia* stand out. For this reason and because their roles in AD have been proposed and elaborated [11], *Malassezia* are discussed next.

## 4. *Malassezia*

*Malassezia* are lipid-requiring basidiomycetes central to the pathophysiology of some forms of AD. Although healthy persons are not usually sensitized to *Malassezia*, many AD patients are, as demonstrated by patch tests, skin prick tests and/or the presence of *Malassezia*-specific IgE [11]. Indeed, of patients with AD, 5–27% of children and 29–65% of adults have *Malassezia*-specific IgE [11]. As most *Malassezia* lack a fatty-acid synthase gene, they require for survival an exogenous source of fatty acids [62]. In the case of species that colonize mammalian skin, the source is sebum. Consistent with this nutritive dependence, *Malassezia* localize to sebum-rich sites, including the scalp, face, chest and upper back [63]. Across these sites, different species of *Malassezia* predominate, e.g., *M. restricta* preferentially colonizes the forehead and external auditory canal whilst *M. globosa* colonizes the back and occiput [64].

Interestingly, IgE against *Malassezia* is associated with a subtype of AD that is limited to the head and neck [11,58,65,66]. One study found IgE against *Malassezia* in 100% of patients with such limited AD, but in only 28% of patients with AD that lacked head and neck involvement [67]. For this reason and because: (1) clinical disease localizes to the sebum-dependent sites of *Malassezia*, and (2) anti-fungal therapy alleviates disease [68], *Malassezia* are claimed cardinal to development of the AD subtype. Inasmuch as complexation of *Malassezia* allergens with IgE can induce inflammation characteristic of AD, it seems reasonable to assume that, in this case, IgE is the primary effector of disease.

*Malassezia* involvement in AD helps account for many other AD-related phenomena. As one example, alkaline soaps elicit AD lesions in susceptible individuals [69]. Alkaline conditions increase expression of *Malassezia* allergens [70]. Relatedly, the *M. globosa* analog of heat shock protein 70 (hsp70) is IgE-reactive in some AD patients [71]. In fungi, alkalinity prompts expression of proteins of the hsp70 family [72]. Thus, it appears manifestations of AD require both habitation by an IgE-reactive microbe and the environment-driven expression of microbe-derived allergens. 

Although *Malassezia* involvement in AD is best supported, it is conceivable that any skin microbe might express proteins capable of eliciting anti-acarian responses. Indeed, IgE against staphylococcal allergens have been demonstrated in AD patients [59,60]. Just as AD of the head and neck can be attributed to IgE against *Malassezia*, other patterns of clinical AD can likely be attributed to other skin flora. Although *Malassezia* are the primary fungal organisms of adults, the fungal organisms of children are much more diverse [73], potentially explaining, at least in part, the patterns of AD seen transiently in children [74].

## 5. *Demodex* and *Dermatophagoides*

Importantly, two genera of acarians, *Demodex* and *Dermatophagoides*, have been demonstrated in increased number on the skin of AD patients [75,76]. *Demodex* spp. (class Arachnida, subclass Acari, order Trombidiformes, family Demodecidae) are ectoparasites specially adapted to inhabit mammalian skin. They reside within the pilosebaceous unit [77]. Over 100 species have been identified, only two of which are commensal human flora: *D. folliculorum*, which resides within hair follicles, and *D. brevis*, which resides within sebaceous glands [78]. *Demodex* derive sustenance from sebum primarily, and from the contents of follicular and glandular epithelial cells [18,79]. Consistent with this nutritive dependence, *Demodex* routinely colonize the face, ears and neck, and less commonly colonize the chest, back, genitals and buttocks [18]. *Demodex* are found within ectopic sebaceous glands of buccal mucosa [80] and within nares [76]. The last two sites provide opportunity for acarians to interface with human dietary and/or inhaled elements. Importantly, cosmetics formulated from human foodstuffs also provide such opportunity [81]. 

*Demodex* colonize both humans and domesticated mammals [82]. Although *Demodex* spp. are mostly monoxenic, human infestation by non-human species occurs [83,84]. Furthermore, DNA of the human mites, *D. folliculorum* and *D. brevis*, have been isolated from privately owned cats [82]. Thus, migration of *Demodex* betwixt pets and humans occurs. Additionally, because *Demodex* feed on epithelial materials of their hosts, interspecies migration likely facilitates generation of specious IgE against cat and/or dog dander. 

There is evidence that the vector activity of *Demodex* spp. contributes to at least one human disease. Rosacea, a chronic, relapsing dermatopathology of the face, is characterized by demodicosis and elevated IgE [85]. Importantly, afflicted patients produce an immune response against antigens derived from the bacterium, *Bacillus oleronius*, isolated from *Demodex* [86]. As the symptoms of rosacea can be mitigated by antibiotics that target, *B. oleronius*, it has been hypothesized that this mite-derived bacterium is cardinal to pathophysiology [87]. 

Phylogenetic analyses of *D*. *folliculorum* populations that inhabit family units and ancestral human lineages are informative. Firstly, the sharing of *D. folliculorum* haplotypes within families indicates that frequent close physical contact leads to transmission of mites. Secondly, ancestral human lineages maintain their mite pedigrees, e.g., African Americans maintain an African mite pedigree whilst European Americans maintain a European mite pedigree [88]. The implication is that mite pedigrees adapt to specific skin traits either because they select them or because they are otherwise unable to survive. It follows, then, that epidermal proteins that affect mite selectivity and/or fitness will necessarily influence the inheritance pattern of allergic disease. 

Like *Malassezia*, *Demodex* subsist on sebum [79], and both species produce lipases that digest triglycerides [89,90]. This nutritive dependence results in co-localization of the two species. In the case of the representative organism, *M. furfur*, lipases are active between pH 4 and 10 [91]. Although lipolysis occurs both intracellularly and extracellularly, alkaline conditions shift processing outside the cell [91]. Such processing, in turn, involves increased secretion of materials relevant to lipid digestion [90]. *Demodex* infestation results in increased pH of afflicted skin [92]. Thus, in an alkaline skin environment of the sort in which the species coexist, sebum consumed by *Demodex* likely contains malassezial elements.

In contrast to *Demodex*, *Dermatophagoides* (class Arachnida, subclass Acari, order Sarcoptiformes, family Pyroglyphidae) are strongly associated with AD and patients with AD produce anti-*Dermatophagoides* antibodies [93]. *Dermatophagoides* do not consume sebum. Their diet consists primarily of mammalian dander. Additionally, they are mycophagous and consume *Alternaria alternate* and *Cladosporium sphaerospermum* [94], well-known contributors of human allergens [95,96]. No study has yet explored the dietary contribution of *Malassezia* to *Dermatophagoides*. Given the presence of the latter on human skin and the paucity of non-malassezial fungal species, ingestion of *Malassezia* by *Dermatophagoides* seems likely. 

## 6. Therapy Reimagined

In keeping with The Acari Hypothesis, AD is a disease process attributable to either: (1) active epithelial infestation by an acarian, or (2) an immunopathological state elicited by an acarian infestation, during which bystander materials were inadvertently targeted. Given either of these, initial management of AD should include identification and elimination of the acarian. Further symptom mitigation would then involve delineation of relevant allergens, including and especially microbes and their associated antigens. Having delineated relevant allergens, clinical disease might best be ameliorated using therapies that minimize allergen burden and/or disrupt relevant immune cascades. Allergen burden might be eliminated by avoiding environmental triggers that prompt microbial expression of relevant allergens (e.g., discontinue use of alkaline soaps) and/or reducing the number of allergen-producing microbes (e.g., treatment with an antibiotic or an antifungal agent). Symptom-inducing immune cascades might be interrupted using: (1) omalizumab, a monoclonal antibody that inhibits the binding of IgE to FcεRI [97,98], (2) tralokinumab and lebrikizumab, monoclonal antibodies that target IL-13 and its signaling [99,100], (3) nemolizumab, a monoclonal antibody that binds the IL-31 receptor and blocks IL-31 signaling [101], and/or (4) dupilumab, an antibody that antagonizes IL-4 at its receptor [102]. 

As targets of IgE, acarians have evolved means to neutralize or evade the effects of IgE activation. Thus, agents derived from the acarian proteome might prove therapeutic. As one example, votucalis, a histamine-binding protein expressed in the saliva of the tick, *Rhipicephalus appendiculatus*, might reduce AD-associated itch and pain [103].

## 7. Connecting Dots

Conceptually, IgE should be regarded as either an anti-acarian immune reactant or as a specious—albeit acarian-triggered—immunopathological reactant. Gross responses elicited by IgE, i.e., itching, spitting, vomiting and defecating, operate as intended to rid epithelial surfaces of what the mammalian immune system recognizes/perceives as vector-active acarians. The prevalence of IgE against synanthropic acarians suggests the vector activity of the organisms must be especially problematic for mammals. Given the narrow functionality of IgE and its conservation across all extant mammalian lineages, it seems likely that one or more acarian species exerted selective pressure critical to the evolution of class Mammalia.

Although this report elaborates a role for acarians in the pathophysiology of AD, other diseases associated with IgE likely also involve acarians. *Demodex* have been implicated in the pathophysiology of recurrent herpetic keratitis [104]. Persons with recurrent herpetic keratitis present with cylindrical dandruff [105], a finding pathognomonic of demodicosis. Importantly, treatment of ocular demodicosis resolves the keratitis [104]. Of note, persons suffering from recurrent herpetic infection have elevated IgE at the time of exacerbation [106]. Given: (1) the association of recurrent herpetic infections with IgE, and (2) the coinciding of anatomical distributions of herpetic lesions and *Demodex* colonization, it is plausible that *Demodex* plays a role in all forms of recurrent herpetic infection.

Chronic obstructive pulmonary disease (COPD) is a relapsing condition characterized by poorly reversible obstruction of pulmonary airways and by pulmonary inflammation [107]. A hallmark of COPD is bronchospasm [108], a feature shared with anaphylaxis and asthma. Smoking is the most important causative factor for COPD [109]. A substantial percentage of COPD patients have elevated IgE and/or eosinophilic infiltrate [110,111], effectors of allergic disease. Tobacco stores contain detectable levels of acarian species, and the mite content of tobacco products correlates with severity of respiratory disease in smokers [112]. Interestingly, workers that process tobacco products have IgE-reactivity to various fungi [113]. Thus, tobacco products may not only be poisonous and carcinogenic [114], but they may also be uniquely immunogenic, a consequence of the infestation of tobacco products by acarian species.

Leprosy, a chronic infectious disease caused by the obligate intracellular bacterium, *Mycobacterium leprae*, involves the skin and peripheral nerves, especially those of the face and hands [115]. Although much is known about the pathophysiology of leprosy, the means of transmission of the disease is poorly understood. The nine-banded armadillo, *Dasypus novemcinctus*, is a reservoir of *M. leprae*, and persons that spend time in the habitat of *D. novemcinctus* are at risk for contracting leprosy, even without directly contacting armadillos [116]. For this reason, it has been suggested acarians are vectors of *M. leprae*. Indeed, at least one tick, *Amblyomma maculatum,* can harbor *M. leprae* and transmit it vertically [117]. *A. maculatum* can also affect horizontal transmission of *M. leprae* to vertebrate hosts [117]. Human-to-human transmission requires prolonged intimate contact, and, within human populations, *Demodex* may serve as bacterial vector [118,119]. Relatedly, some patients infected with *M. leprae* have elevated IgE [120,121], consistent with acarian involvement. 

Although only hypothetical, these examples suggest acarians may be influencing human health in ways not previously considered. The Acari Hypothesis provides a plausible contextual framework with which to test these and other proposals relevant to IgE-mediated diseases, not the least of which is atopic dermatitis.

## Data Availability

Not applicable.

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
