# Peer review of "The Acari Hypothesis, III: Atopic Dermatitis"

_pathogens, 2022, doi:10.3390/pathogens11101083_

Round 1

Reviewer 1 Report

The manuscript presents a hypothesis that arthropods can act as “allergen” vectors” in atopic dermatitis. The authors laid out their ideas is detail, and the hypothesis paper is interesting and inspiring to read. From the view of a parasitologist there are a number of points the authors may want to address to strengthen their perspective and to clarify some points.

First, the term “vector” is usually used for the transmission capacity of arthropods for infectious agents, not for the mechanical transfer of antigens. Of course the term can be used as it is, however, the authors should provide an explanation about their use of this term (I was confused at the beginning because I anticipated a correlation between pathogen transmission and AD)

Second, there are a number of interesting studies on tick saliva constituents and allergy, e.g. the alpha-Gal Syndrome (Sharma SR, Karim S. Tick Saliva and the Alpha-Gal Syndrome: Finding a Needle in a Haystack. Front Cell Infect Microbiol. 2021 Jul 20;11:680264. doi: 10.3389/fcimb.2021.680264.) which may support the general notion that arthropod proteins/glycans play a role in allergy. Ticks are mentioned in a rather unmotivated way at the very end of the manuscript (l. 312), they are very different in lifestyle from Demodex or other mites as they feed on blood which is more or less sterile, so transmission of bacteria, unless in a true host-pathogen-related manner (Anaplasma, Borrelia…) seems not so important compared to mites on/in the superficial skin. The transmission of Mycobacterium leprae by ticks on the other hand is supported by very old literature only, so this appears to be questionable.

Third, the cat allergy. l. 76: allergies against cat dander/hair are not induced by the coat itself and also not by arthropods but in >90% of cases by Fel d1 (https://www.ncbi.nlm.nih.gov/pmc/articles/PMC5891966/) which is found in cat saliva (and is transmitted to the fur by licking; Anti-Fel d1 immunoglobulin Y antibody-containing egg ingredient lowers allergen levels in cat saliva - Ebenezer Satyaraj, Qinghong Li, Peichuan Sun, Scott Sherrill, 2019 (sagepub.com)). T cell epitope specific effects are responsible for the increase in IgE against Fel 1 in AD patients (T Cell Epitope-Specific Defects in the Immune Response to Cat Allergen in Patients with Atopic Dermatitis - ScienceDirect). Allergen on fur can be reduced by feeding anti-Fel d1 chicken IgY in cat feed., and here seems to be a breed disposition for hypo-and hyperallergenic cats.

Fourth, the biology of Demodex is not well described and possibly misunderstood here. For one, the cited literature provides no evidence that Demodex does cross the species border, the paper by Ferreira et al. (citation no. 83) does not describe Demodex in general but a new, so far unnamed species in cats (none of the specimen were assigned to D. folliculorum!), and two citations (84, 85) on cross-species could not be retrieved from the internet (Akbarijo) or did not provide convincing evidence of transmission (Esenkaya). In addition, Demodex in dogs is very well described and D. canis is transmitted exclusively vertically, from the bitch to the puppies, in the first hours after birth in explaining the “family” unit” population structure of the mites. It is assumed that the rate of positive animals is very high but clinical cases are comparatively rare and usually restricted either to juvenile dogs or certain breeds or families. In affected individuals mites proliferate unduely and cause clinical disease, especially in correlation with bacterial secondary infections. In dogs there is no evidence that Demodex infestation is in any way correlated with AD, which is frequently food-associated (and maybe induced by tick, see above).

However, other Demodex species, e.g. D. gatoi in cats, can be transmitted horizontally by direct contact, but these mites do not life in the skin follicle but more superficially in the skin. This may not be so important here, although infested cats display signs of pruritus indicating allergens being involved, while uncomplicated D. canis infestation is usually not accompanied by pruritus (unlike Sarcoptes infestation). Complicated demodicosis in dogs often involved Malassezia infections, and acaridical treatment usually also resolves Malassezia associated otitis (Tarallo VD, Lia RP, Sasanelli M, Cafarchia C, Otranto D. Efficacy of Amitraz plus Metaflumizone for the treatment of canine demodicosis associated with Malassezia pachydermatis. Parasit Vectors. 2009 Mar 5;2(1):13. doi: 10.1186/1756-3305-2-13.). Malassezia pachydermatis is a common commensal on the skin of dogs (Mason IS, Mason KV, Lloyd DH. A review of the biology of canine skin with respect to the commensals Staphylococcus intermedius, Demodex canis and Malassezia pachydermatis. Vet Dermatol. 1996 Sep;7(3):119-132. doi: 10.1111/j.1365-3164.1996.tb00237.x.) and the whole hypothesis could possibly be expanded to animals as well (this is an optional additional aspect to be added unless the authors consider this to be too extensive for the manuscript).

The authors may want to address these points if they think this contribute to the clarity of their hypothesis.

A few formal issues need to be addressed in a revision: l. 45 and elsewhere: line formatting seems to be incorrect. L .116: Spickett, not SPICKETT.

Reviewer 2 Report

You have made an interesting point.

Please find my remarks and questions in the pdf file below.
